# Pharmacokinetic-pharmacodynamic modeling of benznidazole and its antitrypanosomal activity in a murine model of chronic Chagas disease

Frauke Assmus[1,2], Ayorinde Adehin[1,2], Richard M. Hoglund[1,2], Amanda Fortes Francisco[3], Michael D. Lewis[3,4], John M. Kelly[3], Susan A. Charman[5], Karen L. White[5], David M. Shackleford[5], Fanny Escudié[6], Eric Chatelain[6], Ivan Scandale[6], Joel Tarning [1,2]*

**1** Mahidol Oxford Tropical Medicine Research Unit, Faculty of Tropical Medicine, Mahidol University, Bangkok, Thailand, **2** Centre for Tropical Medicine and Global Health, Nuffield Department of Medicine, University of Oxford, Oxford, United Kingdom, **3** Department of Infection Biology, Faculty of Infectious and Tropical Diseases, London School of Hygiene & Tropical Medicine, London, United Kingdom, **4** Division of Biomedical Sciences, Warwick Medical School, University of Warwick, Coventry, United Kingdom, **5** Centre for Drug Candidate Optimisation, Monash University, Melbourne, Australia, **6** Drugs for Neglected Disease initiative, Geneva, Switzerland

* joel@tropmedres.ac

## Abstract

### Background

There is an urgent need for improved treatments for Chagas disease, a neglected tropical infection caused by the protozoan parasite *Trypanosoma cruzi*. Benznidazole, the first line therapy, has severe limitations such as poor tolerability and variable efficacy in the chronic stage of infection. To optimize dosing regimens, a better understanding of the pharmacokinetic/pharmacodynamic (PK/PD) relationship for benznidazole is crucial. This study aimed to characterize the population pharmacokinetic properties of benznidazole in mice and investigate the relationship between exposure and antitrypanosomal activity in *T. cruzi* infected mice.

### Methodology/principal findings

Antitrypanosomal activity was assessed in 118 BALB/c mice with chronic-stage *T. cruzi* infection, utilizing highly sensitive in vivo bioluminescence imaging (BLI). Benznidazole was administered at doses ranging from 10 to 100 mg/kg for 5–20 days. The pharmacokinetic properties of benznidazole were evaluated in 52 uninfected BALB/c mice using nonlinear mixed-effects modeling. The relationship between simulated benznidazole exposure and sterile parasitological cure in the BLI experiments was evaluated by logistic regression and partial least squares – discriminant analysis.

Benznidazole pharmacokinetics in mice were well described by a one-compartment disposition model with first-order absorption, with higher doses associated with

**Data availability statement:** All information, including the underlying code, is available within the manuscript and supporting information.

**Funding:** IS, EC and FE received funding through DNDi from the Bundesministerium für Bildung und Forschung through KfW (Germany), the Foreign Commonwealth & Development Office (FCDO, UK), Directorate-General for International Cooperation (DGIS, Netherlands); Direktion für Entwicklung und Zusammenarbeit (DEZA, Switzerland); and for its overall mission from Médecins Sans Frontières International." should be replaced with: "This work was supported by Drugs for Neglected Diseases initiative (DNDi). DNDi received financial support for this work from Bundesministerium für Bildung und Forschung through KfW (BMBF), Germany, UK Aid, UK, Directorate-General for International Cooperation (DGIS), Netherlands; Swiss Agency for Development and Cooperation (SDC), Switzerland; and for its overall mission from Médecins Sans Frontières International.

**Competing interests:** The authors have declared that no competing interests exist.

slower absorption. Univariate logistic regression revealed a significant correlation between drug exposure and the probability of parasitological cure. Total plasma exposure, time above $IC_{90}$ and peak plasma concentration were all strongly associated with efficacy, provided that benznidazole was administered for at least 5 days.

## Conclusions/significance

This is the first study to successfully quantify the dose-response relationship for benznidazole in *T. cruzi*-infected mice using preclinical BLI data. Our results underscore the complexity of distinguishing PK/PD drivers of efficacy due to high collinearity between PK/PD index parameters, and we propose dose-fractionation studies for future research. Studying the PK/PD relationship using the BLI model provides valuable insights, aiding hypothesis generation through endpoint assessment of parasite infection.

## Author summary

Chagas disease, caused by the parasite *Trypanosoma cruzi*, affects millions worldwide, yet treatments for this neglected tropical disease are far from ideal. Benznidazole, the standard treatment, is poorly tolerated and shows variable efficacy in chronic infections. Optimizing benznidazole dosing regimens requires a better understanding of how its plasma exposure relates to its activity against *T. cruzi* parasites. We used bioluminescence imaging (BLI) to monitor *T. cruzi* infection in mice treated with various benznidazole dosing regimens, allowing us to assess its antiparasitic activity. We also developed a model to describe the drug's pharmacokinetic behavior in mice. Our analyses revealed that higher doses of benznidazole were absorbed more slowly, and drug exposure in plasma—such as total exposure, peak concentrations, and time above a threshold concentration—was strongly linked to the probability of curing the infection. This study is the first to characterize the dose-response relationship for benznidazole in mice using BLI. Our findings highlight the complexity of identifying efficacy drivers due to high collinearity among exposure metrics, suggesting the need for further research. Given the lack of reliable markers for cure in humans, this translational research is crucial for advancing drug development and improving treatment strategies for Chagas disease patients.

## Introduction

Chagas disease (CD), endemic in 21 Latin American countries, is a neglected tropical infection caused by the protozoan parasite *Trypanosoma cruzi* (*T. cruzi*). Approximately 6 to 7 million people worldwide are infected, and a further 75 million are estimated to be at risk [1]. With the rise of migration and globalization, the disease has spread into previously unaffected areas and is now an emerging, global public health concern [2,3].

CD initially presents with an acute phase, which can be asymptomatic or characterized by mild, non-specific symptoms in the majority of cases [1,3]. While most individuals during the chronic phase of infection remain asymptomatic (indeterminate CD), 20 to 30% of patients progress to develop cardiac disorders, with cardiomyopathy standing out as the most severe and often fatal complication. Less frequently, patients develop gastrointestinal disorders such as megacolon and/or megaesophagus [1,3,4].

A century has passed since the discovery of CD, yet only two drugs are currently available for its treatment: nifurtimox and benznidazole, both of which come with severe limitations [5,6]. While demonstrating high efficacy during the acute phase of infection, cure rates during the chronic phase appear to be much more variable among adult patients, decreasing with the duration of infection in chronic symptomatic patients [4,7]. Furthermore, tolerability is poor [8], leading to treatment discontinuation rates ranging from 14.5% to 75% for nifurtimox and 9% to 29% for benznidazole [7]. Therefore, there is an urgent need for new treatment options that offer an improved tolerability and efficacy profile [9].

Drug discovery and development for CD is extremely complex and has been hampered by a limited understanding of disease pathology and host-parasite interactions [9]. Only a few new drugs have progressed through lead optimization into clinical development [9], such as the ergosterol inhibitors posaconazole and fosravuconazole. Unfortunately, results from clinical trials with azoles were disappointing [10–12], highlighting the need for better translational tools to bridge the gap between preclinical and clinical research [13,14].

Recent efforts to improve the treatment of CD patients have primarily focused on optimizing benznidazole treatment regimens, exploring shorter treatment durations, lower doses, intermittent dosing and/or combination therapies [15–19]. However, identifying an optimum treatment regimen presents a major challenge, given the absence of an early biomarker capable of predicting parasitological cure and clinical efficacy in humans [20]. Serology-based techniques require years of follow-up to confirm seroconversion and parasitological cure. While PCR is commonly used in clinical trials to detect *T. cruzi*, it primarily indicates treatment failure rather than success, given the extremely low parasitaemia during the chronic phase [10,11,21]. These technical hurdles in defining a pharmacodynamic (PD) endpoint have also greatly complicated the establishment of pharmacokinetic/pharmacodynamic (PK/PD) relationships [9], essential for guiding dosing optimizations. To date, the relationship between benznidazole dose, plasma exposure, and variability in treatment response remains poorly understood, and the factors driving its efficacy and safety are still unclear.

In view of the challenges in CD drug discovery, animal models have played an integral role as potential translational tools [22]. Particularly, the development of a highly sensitive bioluminescence imaging (BLI) technique has advanced the understanding of CD by allowing real-time, in vivo monitoring of the parasite burden in mice [23]. The BLI technique relies on capturing tissue-penetrating orange-red light emitted by *T. cruzi* parasites expressing a modified firefly luciferase in infected animals. Although BLI models have their own detection limits [24], the technique provides a valuable measure of parasitological cure in benznidazole-treated *T. cruzi* infected animals.

The aim of this study was to leverage the BLI technique to establish the PK/PD relationship for benznidazole in a mouse model of CD. Specifically, our goals were i) to collate benznidazole efficacy data from the highly sensitive BLI method in chronically *T. cruzi* infected mice, ii) to describe the population pharmacokinetic properties of benznidazole in uninfected mice and iii) to quantify the relationship between simulated benznidazole plasma exposure and parasitological cure in the murine model of *T. cruzi* infections.

## Methods

### Ethics statement

Infection experiments were approved by the LSHTM Ethics Committee and performed under UK Home Office licence PPL70/8207. All methods and manipulations for the benznidazole efficacy studies were performed in accordance with the requirements of this licence. All PK studies in non-infected mice conformed to the Australian Code of Practice for the

Care and Use of Animals for Scientific Purposes and were approved by the Monash Institute of Pharmaceutical Sciences Animal Ethics Committee.

## Benznidazole efficacy in mice

Drug efficacy of benznidazole in BALB/c mice chronically infected with *T. cruzi* was assessed with a highly sensitive BLI method as described previously [23]. Briefly, female BALB/c mice 7–8 weeks old were infected with 1000 bioluminescent *T. cruzi* CL Brener blood trypomastigotes derived from a SCID mouse and were treated with benznidazole at the chronic stage of infection via oral gavage [25]. In this model, chronic infection is defined as the stable phase following the acute stage, typically transitioning between days 50–70 post-infection in an immune-mediated process. Beyond this point, the chronic infection remains stable for over a year, as inferred from consistent BLI flux levels (average ~1 × 10⁶ p/s) [23,25].

Mice had access to food and water *ad libitum* and weighed between 20.5 g and 28.1 g at initiation of dosing (median weight 25 g). Benznidazole was prepared as an aqueous formulation containing 0.5% (w/v) hydroxypropyl methylcellulose and 0.4% (v/v) Tween 80. Doses, individually adjusted according to each mouse's weight, ranged from 10 to 100 mg/kg of benznidazole, administered for 5 to 20 days (Table 1, regimens a - j). The sample sizes for each regimen were based on data from multiple drug discovery projects, reflecting the specific requirements and design of the individual experiments. For instance, the 100 mg/kg for 10 days regimen served as a reference across multiple experiments, while other regimens were included to assess new dosing scenarios and explore dose-response relationships. Efficacy data for dosing scenarios c), i) and j) were previously reported [25]. Additional experiments were performed for dosing scenarios a), b), e), f) and g), resulting in an increased sample size. Dosing scenarios d) and h) are reported here for the first time. The efficacy of benznidazole was assessed both in vivo and ex vivo (imaging of dissected organs and tissues). Mice were considered cured if they were bioluminescence negative by both in vivo and ex vivo imaging at the experimental endpoint, following immunosuppression using cyclophosphamide to allow parasite relapse visualization [25].

**Table 1. Summary of benznidazole dosing regimens investigated in *T. cruzi* chronically infected mice (bioluminescence imaging studies), along with simulated median plasma exposures and parasitological cure rates.**

| Dosing regimen[a] | Cumulative dose (mg/kg) | $C_{MAX}$ (µg/mL) | $AUC_{12}$ (µg×h/mL) | $AUC_{24}$ (µg×h/mL) | $AUC_{\infty}$ (µg×h/mL) | $T>IC_{90}$ (days) | N cured mice[b]/ N total mice (%) |
|---|---|---|---|---|---|---|---|
| **a)** 100 mg/kg, 10 days, QD | 1000 | 46.00 | 193.4 | 194.5 | 1945 | 2.86 | 25/ 27 (92.6%) |
| **b)** 100 mg/kg, 5 days, QD | 500 | 46.08 | 195.4 | 196.9 | 985 | 1.44 | 17/ 17 (100%) |
| **c)** 50 mg/kg, 10 days, BID | 1000 | 28.25 | 96.8 | 194.1 | 1946 | 3.95 | 6/ 6 (100%) |
| **d)** 50 mg/kg, 10 days, QD | 500 | 28.11 | 96.9 | 97.2 | 972 | 1.96 | 2/ 3 (66.7%) |
| **e)** 30 mg/kg, 20 days, QD | 600 | 19.07 | 58.3 | 58.6 | 1173 | 2.85 | 23/ 29 (79.3%) |
| **f)** 30 mg/kg, 10 days, QD | 300 | 18.97 | 58.1 | 58.3 | 583 | 1.41 | 9/ 11 (81.8%) |
| **g)** 30 mg/kg, 5 days, QD | 150 | 19.01 | 58.8 | 59.1 | 295 | 0.72 | 0/ 9 (0%) |
| **h)** 20 mg/kg, 10 days, QD | 200 | 13.70 | 38.8 | 38.9 | 389 | 1.01 | 0/ 4 (0%) |
| **i)** 10 mg/kg, 20 days, QD | 200 | 7.67 | 19.5 | 19.6 | 391 | 0.68 | 1/6 (16.7%) |
| **j)** 10 mg/kg, 10 days, QD | 100 | 7.65 | 19.4 | 19.4 | 194 | 0.33 | 0/ 6 (0%) |

**Abbreviations:** QD, once daily; BID, twice daily; $C_{MAX}$, maximum plasma concentrations; AUC area under the concentration-time curve at 12 hours and 24 hours after dosing ($AUC_{12}$, $AUC_{24}$), as well as cumulative AUC at infinity ($AUC_{\infty}$); $T>IC_{90}$, Time above $IC_{90}$ in plasma (6.427 µg/mL, against the amastigote form of *T. cruzi* (Tulahuen strain).

Dotted lines highlight dosing regimens with high (>90%), medium (< 90%, and > 20%) and low efficacy (< 20%).

[a]Dosing schedules indicate the dose per administration. For BID regimens, the total daily dose is the sum of the doses administered twice daily (e.g., 50 mg/kg BID = 100 mg/kg/day).[b]Parasitological cure, defined as absence of bioluminescence signal after in vivo and ex vivo imaging and immunosuppression.

## Pharmacokinetic data from single dose PK studies in satellite mice

Plasma concentration–time data for benznidazole were available from a previously published satellite PK study in non-infected mice (n = 52) [25]. Satellite mice refer to non-infected animals specifically used in pharmacokinetic studies to provide rich sampling and characterize benznidazole's pharmacokinetic behavior. Briefly, female BALB/c mice were administered a single dose of 10 mg/kg (n = 16), 30 mg/kg (n = 18) and 100 mg/kg (n = 18) benznidazole by oral gavage. A fixed dose volume of 0.2 mL per mouse was used for all dosing regimens. Benznidazole formulations were prepared to achieve concentrations of approximately 1 mg/mL, 3 mg/mL, and 10 mg/mL for the 10, 30, and 100 mg/kg dose levels, respectively. The exact dose administered to each mouse was calculated based on the measured concentration of the formulation and the animal's body weight. Formulations contained 5% (v/v) DMSO and a suspension vehicle composed of 0.5% (w/v) hydroxypropyl methylcellulose, 0.5% (v/v) benzyl alcohol and 0.4% (v/v) Tween 80 in Milli-Q water. The formulations were vortexed, sonicated and dosed to mice within 1.5 hours of preparation. Before each individual administration, formulations were actively resuspended to ensure uniform dispersal of benznidazole in the vehicle. Food and drinking water were available *ad libitum* at dosing and up to 6 hours post-dose for all dose groups.

Blood samples were collected at various time points, up to a maximum of 12 hours (10 mg/kg dose group) and 24 hours (30 and 100 mg/kg groups) post-dose. PK samples from 3 - 4 mice were taken at each time point (0.25, 0.5, 1, 2, 4, 4.5, 6, 8, 10, 12, 14, 16, and 24 hours post-dose). PK sampling was sparse, with one to three blood samples obtained from each mouse, either through submandibular bleeding or terminal cardiac puncture into heparinized tubes. Blood samples were centrifuged immediately, supernatant plasma was removed, and stored at -80°C until analysis by UPLC–MS/MS. The lower limit of quantification (LLOQ) for benznidazole in plasma was set to 5 ng/mL. A summary of the bioanalytical method conditions can be found in (S1 Table).

## Population pharmacokinetic analysis

A total of 110 post-dose PK samples from satellite PK studies in uninfected BALB/c mice were collected across three benznidazole dose groups (10, 30, and 100 mg/kg). Two outlying PK samples were identified and excluded from all analysis. The PK data for the 10 mg/kg and 30 mg/kg dose groups at 8 hours and 12 hours after dose, respectively, were censored as all plasma levels beyond these time points were below the LLOQ. For the 100 mg/kg dose group, benznidazole levels were included for the entire PK sampling time frame (up to 24 h). After censoring, 93 PK samples remained in the dataset, of which 90 were above the LLOQ and were included in the final population PK analysis.

The plasma concentration – time profiles were pooled and analyzed simultaneously using nonlinear mixed-effects modeling in NONMEM, v7.4 (Icon Development Solution, Ellicott City, MD, USA). The plasma concentration data were transformed into their natural logarithms prior to analysis. Throughout the model development process, the first-order conditional estimation method with interactions (FOCE - I) was used. Automation and diagnostics were facilitated by the use of Pirana (v2.9.9), Pearl-speaks-NONMEM (PsN, v5.2) and R (v4.2.2). The objective function value (OFV) was used to discriminate between two competing hierarchical models, with the difference in OFV (ΔOFV) being equivalent to a likelihood ratio test. A significant improvement of the structural model was indicated by a decrease in OFV > 3.84 (p < 0.05, 1 degree of freedom).

One-, two-, and three compartment disposition models were explored, as well as different absorption models (first-order absorption with and without lag time, transit compartment absorption with 1 up to 10 transit compartments). Relative bioavailability (F, fixed to unity for the population) was incorporated into the base model to evaluate the inter-individual variability (IIV) in the absorption of benznidazole, and to investigate the impact of covariates on this parameter. IIV in PK parameters was modeled using an exponential error model, and estimated IIV below 10% was fixed to zero in the final model. Residual unexplained variability was implemented as an additive error on the log-transformed observed concentrations (equivalent to an exponential residual error on an arithmetic scale).

The influence of covariates on pharmacokinetic model parameters was evaluated based on biological plausibility, statistical significance, and model performance. Body weight was implemented a priori as an allometric function on clearance (exponent 0.75) and volume of distribution (exponent 1) [26], centered on the median weight of mice in the satellite PK study (19.4 g). Dose (mg/kg) was evaluated as a covariate on clearance and absorption parameters using a stepwise addition (p < 0.05, ΔOFV = −3.84) and elimination (p < 0.001, ΔOFV = −10.83) approach. Linear, exponential and power functions were investigated, with individual doses centered on the median dose (30 mg/kg).

Basic goodness-of-fit diagnostics were used to identify potential model misspecifications and systematic bias. The predictive performance of the final model was evaluated by a visual predictive check (VPC, n = 1000). Parameter precision was obtained using the sampling importance resampling (SIR) procedure [27].

## Simulation of benznidazole exposure in drug efficacy studies

The final population PK model for benznidazole in non-infected satellite mice was utilized to simulate plasma concentration-time profiles for the various dosing regimens investigated in benznidazole efficacy studies. Median plasma concentration profiles for each dosing scenario (n=10) were simulated in NONMEM, implementing the median weight of mice (25 g) used in benznidazole efficacy studies. The following median pharmacokinetic parameters were extracted: peak plasma concentrations ($C_{MAX}$), areas under the plasma concentration-time curves for up to 12 hours and 24 hours after dosing ($AUC_{12}$, $AUC_{24}$), and cumulative AUC at infinity ($AUC_{\infty}$). Additionally, the time above the target concentration in mice ($T>IC_{90}$), determined as $IC_{90,plasma}$ = 24.7 μM (= 6.43 μg/mL), was calculated. This target concentration was derived from the in vitro $IC_{90}$ of benznidazole against the amastigote form of *T. cruzi* (Tulahuen strain) in 3T3 host cells and was corrected for protein binding to assay medium and mice plasma. $IC_{90,plasma}$ corresponds to the total concentration of benznidazole in mice plasma needed to achieve a 90% reduction in *T. cruzi* amastigote infection. Further details regarding the determination of in vitro antitrypanosomal activity and correction for protein binding are available (**S1 Text**).

## Exposure – response analysis

The relationship between simulated drug exposure and cure (BLI negative after both in vivo and ex vivo imaging) was assessed using binary univariate logistic regression modeling in R (v4.2.2). The logistic regression model is given by:

$$logit(p) = ln\left(\frac{p}{(1-p)}\right) = \beta_0 + \beta_1 x + \varepsilon$$

(1)

where *p* represents the probability and logit (*p*) the log odds of parasitological cure. In Equation 1, $\beta_0$ is the intercept, $\beta_1$ is the coefficient of the predictor variable x, and ε denotes the residual error. Various simulated exposure variables (x) were investigated as predictors of parasitological cure, including median $C_{MAX}$, $AUC_{12}$, $AUC_{24}$, $AUC_{\infty}$ and $T>IC_{90}$. These exposure variables were derived from simulations of the different benznidazole exposures across the various dosing regimen groups, as described above. The goodness of fit diagnostics for the PK/PD index parameters were compared based on the Akaike and Bayesian information criterion (AIC, BIC) as well as McFadden's Pseudo $R^2$.

The predictive performances of the logistic regression models were assessed across the entire range of classification thresholds using the areas under the Receiver Operating Characteristics (ROC) curve. The default classification threshold was set to 50%, classifying mice as cured if the predicted probability of cure exceeded 50%, and all other values as not cured. Additionally, classification thresholds were optimized to maximize the Matthews Correlation Coefficient (MCC), a metric considered more reliable for imbalanced datasets compared to accuracy and F1-score [28]. Notably, optimization towards maximum accuracy and/or maximum MCC yielded the same cutoff values in this study. Additional classification performance metrics, including sensitivity, specificity, and F1 score, were also calculated. Detailed information on these metrics is provided in (S2 Text).

The relationship between drug exposure and parasitological cure was further investigated using partial least squares – discriminant analysis (PLS-DA), performed in R with the 'mdatools' package [29]. This analysis aimed to explore a combination of PK/PD index parameters as potential predictors of parasitological cure, with PLS-DA employed to reduce the multidimensional space. Briefly, PLS-DA is a multivariate dimensionality-reduction method that identifies latent variables by maximizing the covariance between predictor variables and class membership (cured/not cured) [30]. Further details about the PLS-DA model development are provided in (S3 Text).

## Results

### Benznidazole efficacy in *T. cruzi* infected mice

Antitrypanosomal activity of benznidazole against chronic stage *T. cruzi* infections was available from a total of 118 mice and 10 dosing regimens. The dataset comprised literature data (n = 62 mice) [25] and additional experimental results (n = 56 mice), resulting in an extended dataset. The investigated treatments covered a wide range of doses (10, 20, 30, 50, and 100 mg/kg benznidazole) and treatment durations (5, 10, and 20 days). In most cases, benznidazole was administered once daily, with the exception of one regimen involving twice-daily dosing. A summary of benznidazole efficacy in the *T. cruzi* CL Brener-BALB/c model for the various dosing regimens is provided in Table 1, alongside simulated secondary PK/PD index parameters (for details see below).

In brief, parasitological cure was achieved in over 90% of chronically infected mice when treated once daily with 100 mg/kg benznidazole for 5 days (17/17 mice) or 10 days (25/27 mice; scenarios a and b). Similarly, twice daily dosing with 50 mg/kg per administration (100 mg/kg/day) for 10 days cured all mice (6/6; scenario c). In contrast, once daily dosing with 50 mg/kg for 10 days cured only 66.7% of infected mice (2/3; scenario d). Pronounced variability in drug efficacy was also observed with 30 mg/kg benznidazole: cure rates reached approximately 80% in mice treated for 10 days (9/10; scenario f) or 20 days (23/29 mice; scenarios e), while a shorter treatment duration of 5 days failed to cure any mice (0/9; scenario g). Similar non-curative outcomes were observed for lower benznidazole doses (10 and 20 mg/kg, scenarios h-j). With these dosing schedules, a reduction in parasite burden was achieved, but none (0/6; 10 days) or only 16.7% of mice (1/6; 20 days) were ultimately cured.

### Pharmacokinetics of benznidazole

The pharmacokinetic properties of benznidazole were characterized using PK data from 52 female, non-infected BALB/c mice. Each mouse received a single oral dose of 10 mg/kg (n = 16), 30 mg/kg (n = 18), or 100 mg/kg (n = 18) benznidazole. Sparse PK sampling was employed, with 1 to 3 blood samples collected per mouse. In total, 90 plasma concentrations above the LLOQ were included in the final population PK analysis. At the initiation of dosing, mice weighed between 18.1 g and 21.6 g (median weight ± SD: 19.4 g ± 0.92 g).

Benznidazole concentration-time profiles in this satellite PK study were best described by a one-compartment disposition model with first-order absorption and elimination. While a two-compartment disposition model significantly improved the model fit (ΔOFV ≈ -15), it exhibited instability with high variability (> 100% IIV) and low precision (> 70% RSE) in the additional parameters. Moreover, no significant improvement was observed for the two-compartment compared to the one-compartment model when using the M3 BQL approach [31], suggesting that the apparent improvement of a multi-phasic model was driven by LLOQ data. Thus, the one-compartment disposition model was carried forward. More complex absorption models, including a transit absorption model and incorporation of a lag time, were investigated, but this yielded no significant improvement of the model fit. The impact of allometric scaling on the model fit was negligible (ΔOFV = 1.1), given the narrow distribution of body weights in the satellite PK study. Nonetheless, body weight was included a priori as an allometric function on CL/F (exponent = 0.75) and $V_c$/F (exponent = 1.0) to improve the translational value of the model and to allow for simulating exposure parameters in the efficacy experiment with different body weights.

In the present study, a higher dose was associated with a significantly slower absorption rate (ΔOFV = -31.3), as indicated by a decrease in $K_A$ (power function). At the lowest, medium and highest dose, $K_A$ was estimated as 5.11, 2.18 and 0.86 h$^{-1}$, respectively. Additionally, there was a trend towards lower clearance at higher benznidazole doses. It is important to note that the effect size was sensitive to a single time point (8 hours post-dose at 10 mg/kg), where measured benznidazole levels were close to the LLOQ. Given the uncertainty and lack of significance after excluding the 8 h/10 mg/kg data (p > 0.001), the effect of dose on CL/F was not included in the final PK model. The estimated IIVs associated with F and $K_A$ were <10% and therefore not retained in the final model.

Parameter estimates for the final model, along with their relative standard errors (RSEs) and 95% confidence intervals (CIs), are presented in Table 2. Sampling importance resampling (SIR) indicated a robust PK model with moderate-to-high precision in estimating PK parameters and covariate effects (RSE < 18% for all parameters). The final model described the observed concentration-time profiles well with no major model misspecification, as demonstrated in the basic goodness-of-fit plots (S1 Fig). Moreover, the VPC indicated good predictive performance of the final model (Fig 1). It is important to note that non-random censoring of data below the LLOQ can complicate the interpretation of prediction-corrected VPCs [32]. In our study, PK data were censored at 8 hours (10 mg/kg) and 12 hours (30 mg/kg) post-dose, while no censoring was applied in the 100 mg/kg dose group. Therefore, the non-prediction-corrected VPC is presented (Fig 1). For completeness, the prediction-corrected VPC is provided in (S2 Fig). The NONMEM code of the final PK model is provided in (S1 Code).

Secondary PK parameter estimates for each dosing group can be found in (S2 Table). Times to maximum concentration ($T_{MAX}$) ranged between 0.49 hours (10 mg/kg), 0.86 hours (30 mg/kg) and 1.47 hours (100 mg/kg). $C_{MAX}$ ranged between 8 μg/mL (10 mg/kg), 19 μg/mL (30 mg/kg) and 44.7 μg/mL (100 mg/kg). In contrast to $C_{MAX}$, total exposure was nearly dose-linear, with AUC$_\infty$ ranging between 18.1 μg×h/mL (10 mg/kg), 57.3 μg×h/mL (30 mg/kg) and 186 μg×h/mL (100 mg/kg). The terminal elimination half-life for benznidazole was approximately 1.3 h.

### Simulated benznidazole exposure in drug efficacy studies

The final population PK model derived from satellite PK data was employed to simulate the median plasma concentration-time profiles for benznidazole in the different treatment regimens evaluated in benznidazole efficacy studies (Fig 2). Secondary PK/PD index parameters were extracted and are detailed in Table 1, along with the corresponding parasitological cure rates.

**Table 2. Parameter estimates of the final population pharmacokinetic model for benznidazole in uninfected BALB/c mice.**

| Parameter[a] | Population estimate[b] (%RSE)[c] | 95% CI[c] | IIV, %CV[b] (%RSE)[c] | 95% CI[c] |
|---|---|---|---|---|
| *Pharmacokinetics* | | | | |
| $K_A$ (h$^{-1}$) | 2.18 (15.9%) | 1.65 – 3.03 | – | – |
| CL/F (mL/h) | 10.6 (5.6%) | 9.62 – 11.88 | 17.4 (17.0%) | 10.5 – 22.5 |
| $V_c$/F (mL) | 19.7 (8.2%) | 17.05 – 23.18 | 16.6 (29.7%) | 6.9 – 25.6 |
| σ | 0.122 (10.9%) | 0.087 – 0.188 | | |
| *Covariates* | | | | |
| $\theta_{Dose}$ (dose effect on $K_A$) | -0.775 (13.3%) | -0.977 – -0.583 | – | – |

[a]**Abbreviations:** $K_A$, absorption rate constant; CL/F, apparent elimination clearance; $V_c$/F, apparent central volume of distribution; σ, unexplained residual variance (log scale); Population estimates are given for mice weighing 19.4 g (median weight). Allometric scaling on clearance (exponent fixed to 0.75) and volume of distribution (exponent fixed to 1). The dose effect on $K_A$ was described by a power function with dose (centered on the median dose of 30 mg/kg) raised to an estimated exponent: $\left(\frac{Dose_i}{Dose_{median}}\right)^{\theta_{Dose}}$.

[b]Population mean parameter estimates and inter-individual variability (IIV) calculated by NONMEM. The coefficient of variation (% CV) for the IIV was calculated as $100 \times \sqrt{\exp(\omega^2) - 1}$.

[c]Relative standard errors (RSE, %) and 95% confidence intervals (95% CIs) were calculated based on sampling importance resampling (SIR) of the final pharmacokinetic model.

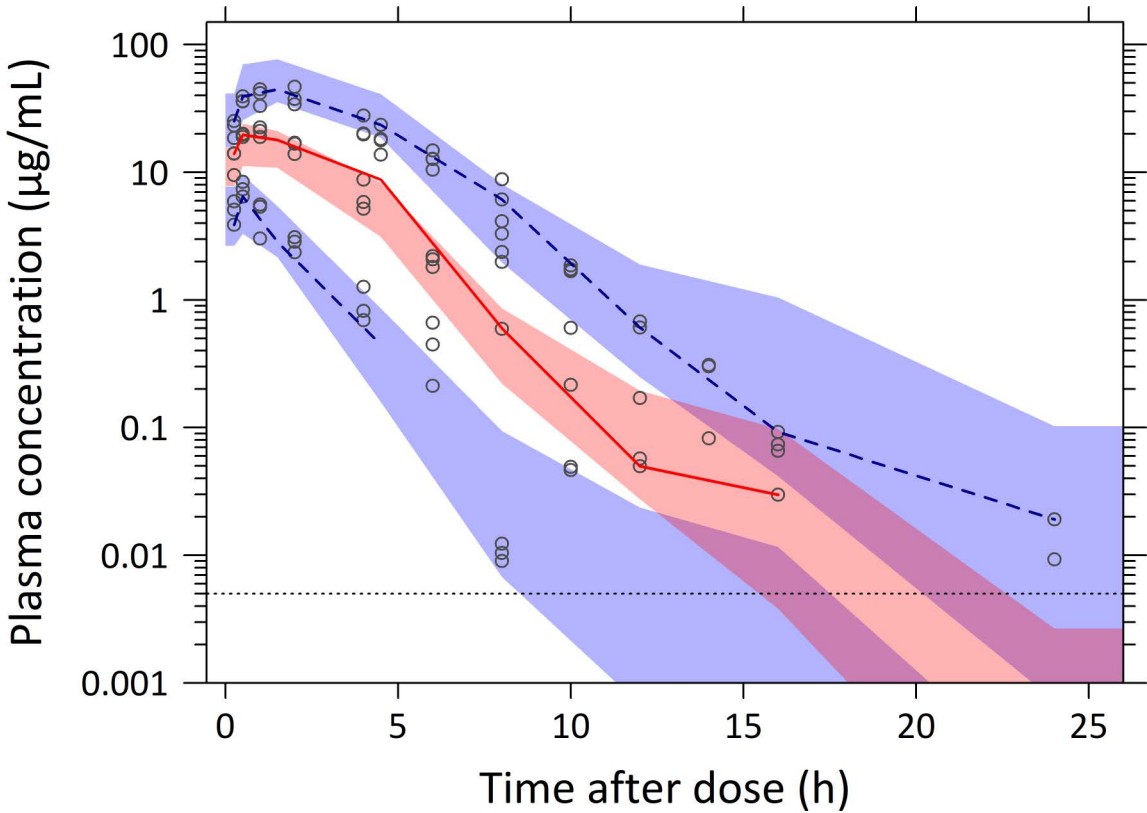

**Fig 1. Visual predictive check of the final population pharmacokinetic model for benznidazole (not prediction-corrected).** Open circles represent observed plasma benznidazole concentrations from 52 female BALB/c mice across all dose groups (10, 30, and 100 mg/kg), including 90 plasma concentrations above the lower limit of quantification (LLOQ). The solid red line represents the 50th percentile (median) and the dashed blue lines represent the 5th and 95th percentiles of the observed data. The shaded areas represent the 95% confidence intervals around the simulated 5th, 50th, and 95th percentiles. The horizontal dashed line represents the LLOQ (5 ng/mL = 0.005 μg/mL).

Simulations demonstrated that higher cumulative doses, resulting in increased total plasma exposure ($AUC_\infty$), were associated with longer durations above the target concentration ($IC_{90,plasma}$= 6.43 μg/mL) against *T. cruzi* amastigotes. The median $AUC_\infty$ for benznidazole in plasma ranged from 194 to 1945 μg×h/mL across the different dosing regimens, representing a 10-fold range. Median $C_{MAX}$ values, ranging from 7.65 to 46 μg/mL, consistently exceeded the target concentration (6.43 μg/mL), regardless of the dosing regimens. Median cumulative $T>IC_{90}$ varied widely, ranging from 0.33 days to 3.95 days, and were highly sensitive to the $IC_{90,plasma}$ (see sensitivity analysis below). Trough concentrations ($C_{trough}$) approached zero before the next dose for both once-daily and twice-daily regimens and were therefore not considered further.

It is noteworthy that strong correlations between PK/PD index parameters were found (**Table 3**), revealing two distinct clusters (see also S3 Fig): i) a pronounced cross-correlation between dose per dosing occasion, daily dose, $C_{MAX}$, $AUC_{12}$ and $AUC_{24}$ (Pearson correlation coefficient, $R^2 > 0.95$). Additionally, ii) a high cross-correlation was observed between cumulative dose, $AUC_\infty$ and $T>IC_{90}$ ($R^2 = 1$ for $AUC_\infty \sim$ cumulative dose; $R^2 = 0.88$ for $AUC_\infty \sim T>IC_{90}$).

### Exposure – parasitological response analysis

**Univariate logistic regression.** To characterize the relationship between benznidazole exposure and efficacy in *T. cruzi* infected mice, a binary univariate logistic regression modeling approach was employed. The distributions of PK/

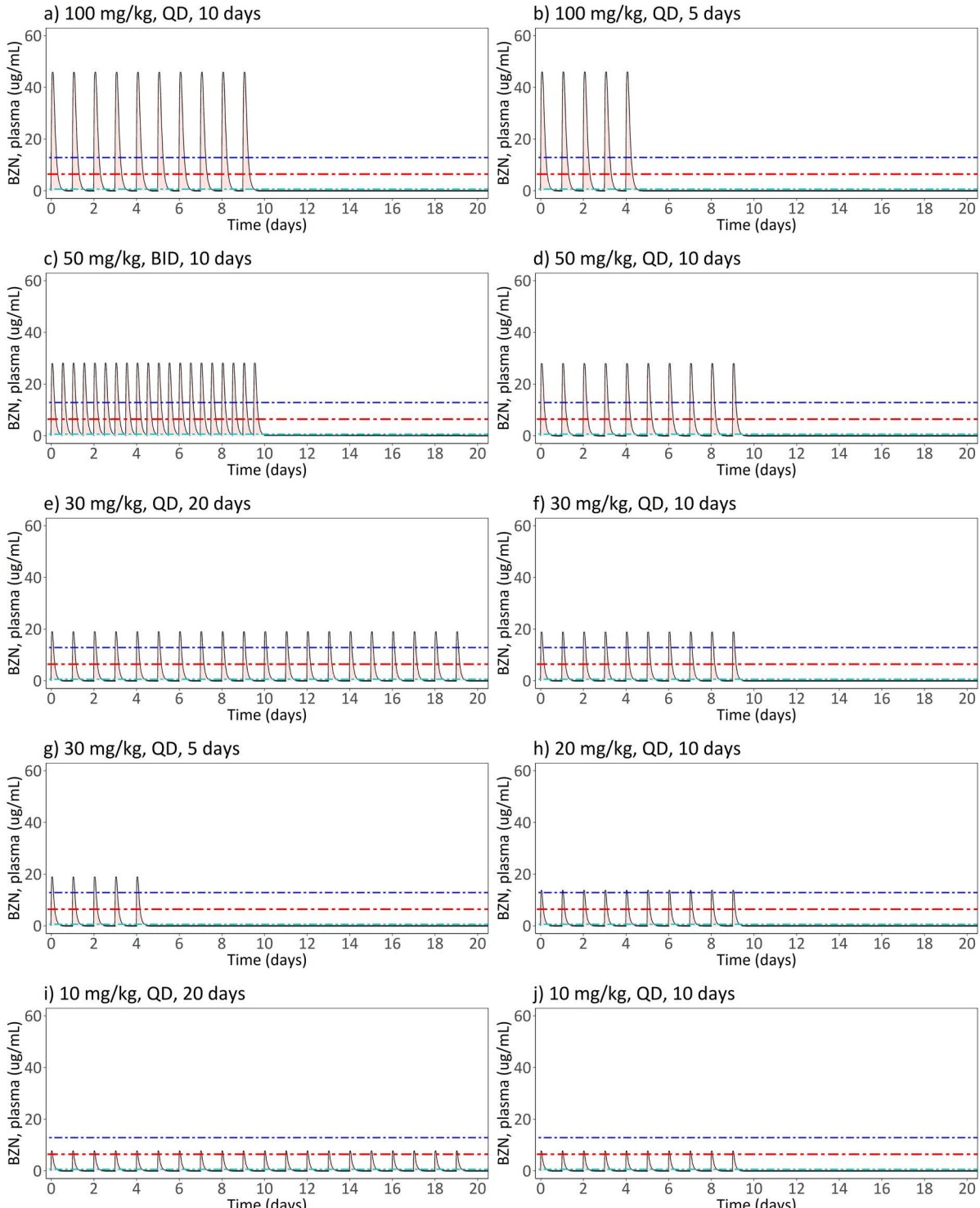

**Fig 2. Simulations of pharmacokinetic median profiles for benznidazole (BZN) according to different dosing regimens investigated in *T. cruzi* infected mice.** Simulations were based on the final population pharmacokinetic model and a 25 g mouse (median weight of mice in drug efficacy studies). Black solid lines represent the median of the simulated benznidazole plasma concentrations over time. The horizontal red line represents the $IC_{90,plasma}$ against the amastigote form of *T. cruzi* (Tulahuen strain). The dark blue and light blue horizontal lines represent two-fold higher and two-fold lower target concentrations, respectively.

Table 3. Correlation matrix of plasma exposure variables for benznidazole, simulated for dosing regimens investigated in bioluminescence imaging studies in *T. cruzi* chronically infected mice. Correlation was assessed using Pearson correlation coefficients (https://cran.r-project.org/web/packages/corrtable/index.html). All coefficients were highly significant (p value < 0.001); $C_{MAX}$, maximum plasma concentrations; AUC area under the concentration-time curve for up to 12 hours and 24 hours after dosing ($AUC_{12}$, $AUC_{24}$), and cumulative AUC extrapolated to infinity ($AUC_\infty$); $T>IC_{90}$, Time above $IC_{90}$ in plasma.

| Dosing regimen | Dose per occasion | Dose per day | Cumulative dose | $C_{MAX}$ | $AUC_{12}$ | $AUC_{24}$ | $AUC_\infty$ | $T>IC_{90}$ |
|---|---|---|---|---|---|---|---|---|
| Dose per occasion | 1 | | | | | | | |
| Dose per day | 0.95 | 1 | | | | | | |
| Cumulative dose | 0.68 | 0.75 | 1 | | | | | |
| $C_{MAX}$ | 1.00 | 0.96 | 0.70 | 1 | | | | |
| $AUC_{12}$ | 1.00 | 0.95 | 0.67 | 1.00 | 1 | | | |
| $AUC_{24}$ | 0.96 | 1.00 | 0.75 | 0.96 | 0.96 | 1 | | |
| $AUC_\infty$ | 0.68 | 0.75 | 1.00 | 0.70 | 0.68 | 0.75 | 1 | |
| $T>IC_{90}$ | 0.32 | 0.44 | 0.88 | 0.36 | 0.32 | 0.43 | 0.88 | 1 |

PD index parameters among mice (not cured vs. cured) are presented in **Fig 3** (upper panel), along with the predicted probabilities of parasitological cure (**Fig 3**, lower panel).

A summary of logistic regression model diagnostics is provided in **Table 4**. For all PK/PD index parameters, a significant correlation was found between simulated benznidazole exposure in plasma and the odds of parasitological cure (p < 0.001). For example, a 1 µg×h/mL increase in $AUC_\infty$ was associated with a 0.33% (95% CI: 0.22% – 0.47%) increase in the odds of parasitological cure. Given the high correlation between cumulative dose and $AUC_\infty$ in this dataset, the performance of the logistic regression model was identical for both metrics. However, we focus on plasma exposure metrics (such as $AUC_\infty$ or $C_{MAX}$) due to their greater potential translational value for human dose predictions.

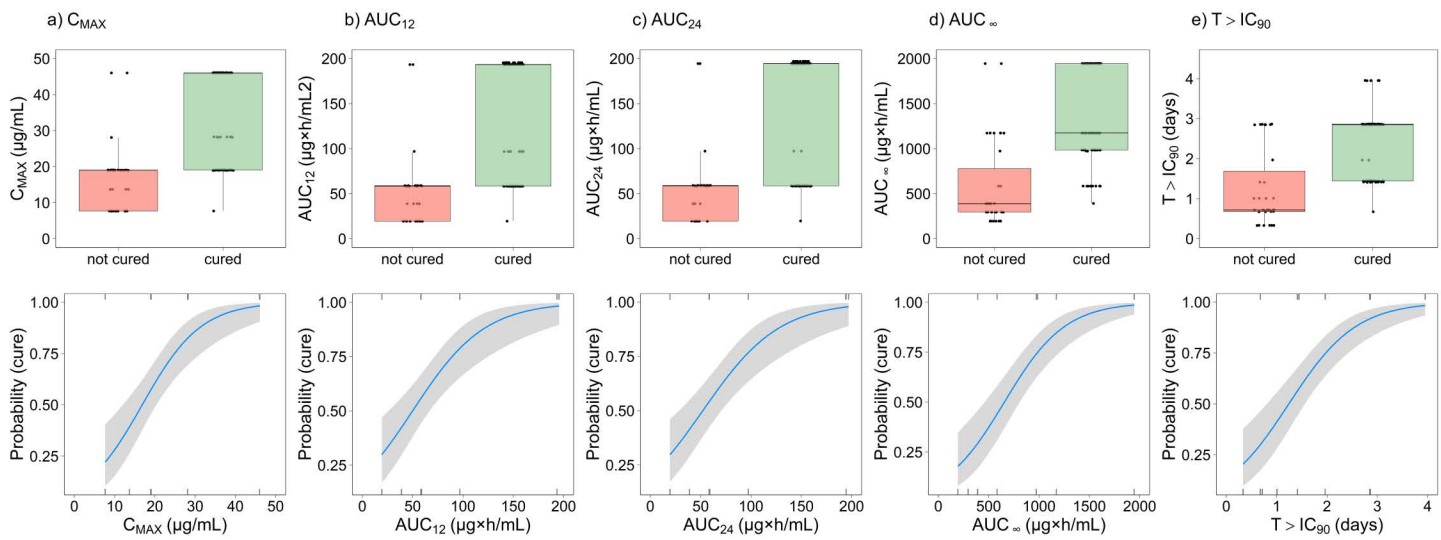

**Fig 3. Relationship between benznidazole exposure and antitrypanosomal activity in *T. cruzi* infected mice.** Boxplots (upper panel) showing the distribution of (a) $C_{MAX}$, (b) $AUC_{12}$, (c) $AUC_{24}$, (d) $AUC_\infty$ and (e) $T>IC_{90}$ for *T. cruzi* infected mice that were not cured (n=35) and cured (n=83). The midline indicates the median, the box corresponds to the interquartile range, and the whiskers extend up to 1.5 times the interquartile range. The lower panel shows corresponding predicted probabilities of antitrypanosomal activity, based on the logistic regression modeling. The solid blue line indicates the median and the shaded area the 95% CI around predicted probabilities.

Goodness-of-fit diagnostics (AIC, BIC, Mc Fadden $R^2$) indicated slightly stronger explanatory power for $AUC_\infty$ in binary regression compared to $C_{MAX}$, $AUC_{12}$, $AUC_{24}$, and $T>IC_{90}$. However, the differences in goodness of fit among the various PK/PD index parameters were generally small, as indicated by ΔAIC less than 10.

ROC curves, displaying sensitivity and (1-specificity) across the entire range of classification thresholds, are presented in **Fig 4**. The areas under the ROC curves for $C_{MAX}$, $AUC_\infty$ and $T>IC_{90}$ all exceeded 85%, with overlapping 95% CIs (**Table 4**), demonstrating good discriminative power for each of the univariate logistic regression models, provided that benznidazole was administered for at least 5 days.

In terms of classification predictions, model performances were sensitive to the choice of the classification threshold. The disparities between the default threshold and the optimized threshold were small, as detailed in (**S3 Table**). For both thresholds, sensitivity (ranging from 88% to 99%) was consistently higher than specificity (ranging from 43% to 74%). When utilizing an optimized probability threshold, model performances based on $AUC_\infty$ or $T>IC_{90}$ correctly classified 90% of mice, while logistic regression with $C_{MAX}$, $AUC_{12}$ and $AUC_{24}$ showed slightly lower accuracies (ranging from 82% to 84%, **Table 4**). Additional overall model performance metrics, such as F1-score and MCC, demonstrated a similar trend, with small differences between model predictions with $AUC_\infty$, $T>IC_{90}$, and $C_{MAX}$.

The developed logistic regression models were used to determine exposure levels required for clearing *T. cruzi* infections in mice (**Table 5**). For example, an increase in $AUC_\infty$ from approximately 1300 µg×h/mL to 1500 µg×h/mL was associated with a median increase in the probability of cure from 90% to 95%. Further increase in $AUC_\infty$ to approximately 2000 µg×h/mL was predicted to result in a 99% probability of cure. When using $T>IC_{90}$ as the efficacy predictor, logistic regression modeling indicated that achieving parasitological cure with 95% and 99% probability requires a minimum duration of 3.2 and 4.3 days above the target concentration, respectively. For $C_{MAX}$, approximately 50 µg/mL benznidazole exposure in plasma is required for a 99% probability of parasitological cure.

**Table 4. Binary univariate logistic regression model diagnostics for the correlation between benznidazole exposure and the odds of parasitological cure in mice chronically infected with *T. cruzi*.**

| Parameter | Cumulative dose (mg/kg) | $C_{MAX}$ (µg/mL) | $AUC_{12}$ (µg×h/mL) | $AUC_{24}$ (µg×h/mL) | $AUC_\infty$ (µg×h/mL) | $T>IC_{90}$ (days) |
|---|---|---|---|---|---|---|
| Intercept (SE)[a] | -2.157 (0.565) | -2.312 (0.666) | -1.394 (0.489) | -1.375 (0.457) | -2.170 (0.566) | -1.859 (0.521) |
| LogOdds (SE)[a] | 0.006 (0.001) | 0.137 (0.032) | 0.028 (0.007) | 0.026 (0.006) | 0.0033 (0.0006) | 1.497 (0.289) |
| Odds (%)[b] | 0.65 | 14.70 | 2.83 | 2.67 | 0.33 | 347 |
| *Goodness of fit* | | | | | | |
| AIC | 99.1 | 105.1 | 108.4 | 105.6 | 98.8 | 108.5 |
| BIC | 104.6 | 110.7 | 113.9 | 113.9 | 104.4 | 114.0 |
| Mc Fadden $R^2$ | 0.338 | 0.295 | 0.273 | 0.292 | 0.339 | 0.272 |
| *Classification performance*[c] | | | | | | |
| Accuracy | 0.90 | 0.84 | 0.82 | 0.82 | 0.90 | 0.90 |
| Sensitivity | 0.99 | 0.88 | 0.99 | 0.99 | 0.99 | 0.99 |
| Specificity | 0.69 | 0.74 | 0.43 | 0.43 | 0.69 | 0.69 |
| F1 score | 0.93 | 0.88 | 0.89 | 0.89 | 0.93 | 0.93 |
| Matthews Correlation Coefficient | 0.753 | 0.556 | 0.556 | 0.556 | 0.753 | 0.753 |
| ROC (%), (95% CI) | 85.8 (77.2 – 94.3) | 87.0 (80.3 – 93.7) | 79.7 (71.3 – 88.0) | 79.9 (71.6 – 88.2) | 86.4 (78.0 – 94.7) | 85.3 (76.8 – 93.8) |

**Abbreviations:** SE, standard error; AIC, Aikaike information criterion; BIC, Bayesian information criterion; ROC, area under the Receiver Operating Characteristics curve.

[a]All values were significant with a p value < 0.01.

[b]Increase in odds of parasitological cure per unit increase in cumulative dose (mg/kg), $C_{MAX}$ (µg/mL), AUC (µg×h/mL), and $T>IC_{90}$ (days).

[c]Accuracy, sensitivity, specificity, F1 score and Matthews Correlation Coefficient were calculated based on optimized probability thresholds.

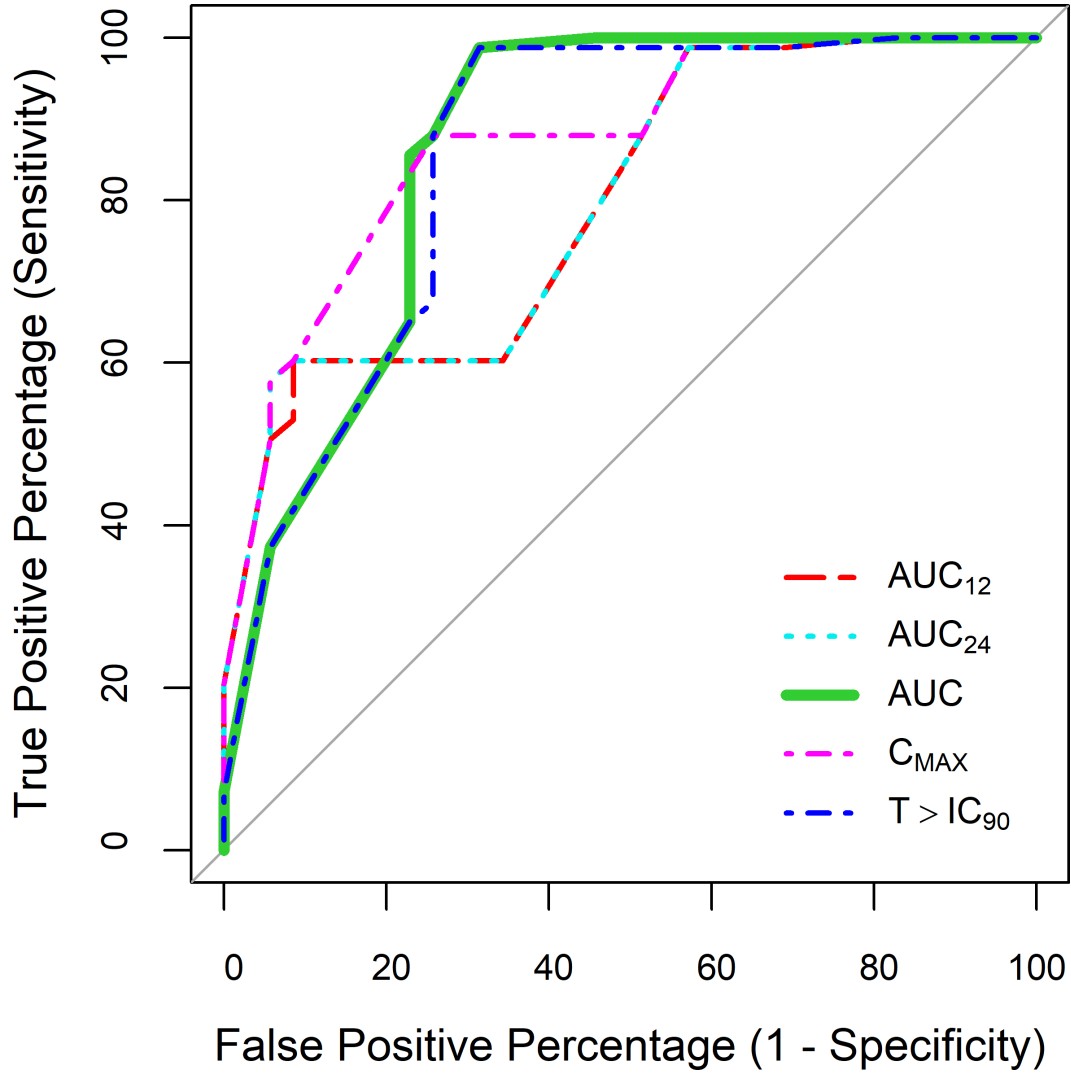

**Fig 4. Receiver Operating Characteristics (ROC) curves for logistic regression, evaluating the relationship between benznidazole plasma exposure and parasitological cure in mice.** The lines represent ROC curves for different exposure metrics: area under the concentration-time curve at 12 hours ($AUC_{12}$) and 24 hours ($AUC_{24}$) after dosing, cumulative AUC to infinity ($AUC_{\infty}$), maximum plasma concentration ($C_{MAX}$), and time above the target concentration ($T>IC_{90}$). The areas under the ROC curves are summarized in Table 4.

**Sensitivity analysis.** A sensitivity analysis was conducted for a range of $IC_{90,plasma}$ values, spanning a 10-fold range. This analysis takes into account that the in vitro $IC_{90}$ is only an approximation of in vivo antitrypanosomal activity in *T. cruzi* infected mice. Times above target concentrations corresponding to different $IC_{90,plasma}$ values are provided in S4 Table, along with binary logistic regression diagnostics (S5 Table). Briefly, the median $T>IC_{90}$ was found to be highly sensitive to variations in $IC_{90,plasma}$. Moreover, there was a strong correlation between $T>IC_{90}$ and $AUC_{\infty}$, particularly when $IC_{90,plasma}$ was assumed to be 2-fold higher than the in vitro value ($R^2 = 0.96$). Consequently, logistic regression models yielded similar model performances when using $AUC_{\infty}$ and $T>IC_{90}$ as predictors of sterile cure in *T. cruzi* infected mice, provided that the $IC_{90,plasma}$ was scaled 2-fold. These models showed overlapping 95% CIs for areas under the ROC curves with $C_{MAX}$, indicating comparable classification performance.

**Table 5. Predicted probabilities of parasitological cure in mice chronically infected with _T.cruzi_, and required benznidazole exposures in plasma.**

| Predicted probability of cure in mice | Cumulative dose (mg/kg) ± SE | AUC$_\infty$ (µg×h/mL) ± SE | T>IC$_{90}$ (days) ± SE | C$_{MAX}$ (µg/mL) ± SE |
|---|---|---|---|---|
| 50% | 335 ± 41.8 | 656 ± 81.5 | 1.24 ± 0.18 | 16.9 ± 1.9 |
| 90% | 676 ± 66.2 | 1319 ± 128 | 2.71 ± 0.27 | 32.9 ± 3.6 |
| 95% | 792 ± 85.3 | 1545 ± 165 | 3.21 ± 0.35 | 38.3 ± 4.8 |
| 99% | 1048 ± 131 | 2044 ± 255 | 4.31 ± 0.55 | 50.4 ± 7.5 |

SE: standard error

**Multivariate analysis.** To explore the potential improvement of model performance through a combination of PK/PD index parameters, a multivariate model for predicting parasitological cure in _T. cruzi_ infected mice was developed. Given the high collinearity among predictor variables, PLS-DA was employed to reduce the multidimensional space. Details about the PLS-DA model evaluation can be found in (S3 Text). The final PLS-DA model, constructed with one latent variable, identified AUC$_\infty$, T>IC$_{90}$, and C$_{MAX}$ as the most relevant predictors for class discrimination, with variable influence on projection (VIP) scores of 1.47, 1.40, and 1.37, respectively. In terms of classification performance, there was no significant improvement between the multivariate PLS-DA model and univariate regression analysis with either AUC$_\infty$ or T>IC$_{90}$. These models yielded identical values for accuracy, sensitivity, and specificity when optimized classification thresholds were applied (**Figure E in** S3 Text, panel b-d). In terms of areas under the ROC curve, no significant differences were found between multivariate and univariate models, regardless of whether AUC$_\infty$, T>IC$_{90}$, or C$_{MAX}$ was applied (**Table B in** S3 Text).

## Discussion

Current benznidazole dosages and treatment protocols rely on scarce data from experimental and non-randomized studies conducted decades ago [33–35], and their appropriateness has been a subject of intense debate [15–19,36]. As benznidazole dosing regimens are re-evaluated, a critical need for a comprehensive understanding of dose-exposure-response relationships has emerged to facilitate the optimization process [9].

This study presents a comprehensive analysis of the relationship between benznidazole dose, plasma exposure, and its antitrypanosomal activity in mice chronically infected with _T. cruzi_. To our knowledge, it is the first PK/PD analysis of benznidazole in mice considering an extended set of PK/PD index parameters as predictors of parasitological cure, focusing on plasma exposure rather than just dose. Furthermore, our analysis employs a robust statistical framework to quantitatively characterize dose-exposure-response relationships, providing valuable insights into benznidazole's pharmacokinetic properties in relation to its antitrypanosomal efficacy.

### Pharmacokinetic properties of benznidazole

Plasma concentration-time profiles of benznidazole in uninfected mice were well described by a one-compartment disposition model with first-order absorption, consistent with previous analyses [37–39]. For example, Perin et al. [37] conducted a population PK analysis of benznidazole in Swiss mice, also identifying a one-compartment model, reflecting wide distribution of benznidazole across organs [37]. Although tissue-to-whole-plasma ratios varied among tissues, they typically remained below 100%, suggesting limited tissue distribution [40,41]. This aligns with the moderate apparent volume of distribution ($V_C$/F = 19.7 mL) observed in our study and described previously [38,39,41]. Higher values for both $V_C$/F (116 mL) and CL/F (48 mL/h) in mice were reported by Perin et al. [37], potentially due to differences in benznidazole formulation and strain-specific variations in oral bioavailability.

Benznidazole, which is poorly soluble in aqueous solution [42,43], showed dose-dependent absorption kinetics in our study, likely due to prolonged dissolution in gastrointestinal fluids. Administering benznidazole as a suspension at doses of 30 mg/kg and 100 mg/kg resulted in delayed absorption ($T_{MAX}$ = 0.86 h and 1.47 h, respectively) compared to the clear 10 mg/kg solution ($T_{MAX}$ = 0.49 h). As a consequence, $C_{MAX}$ values increased less than dose-proportionally, which was not observed by Perin et al. [37]. Nevertheless, $T_{MAX}$ values reported by Perin et al. ($T_{MAX}$ = 0.83 h at 100 mg/kg) and other studies were in close agreement with our results, indicating relatively rapid absorption at these dose levels [37,39–41].

In contrast to $C_{MAX}$, total exposure ($AUC_{\infty}$) showed nearly dose-proportional behavior in our study. The elimination half-life was consistent across different doses ($t_{1/2}$ = 1.3 h) and closely resembled the findings of Workman et al. in BALB/c mice at doses up to 78 mg/kg ($t_{1/2}$ ~ 1.5 h) [40]. Variations in reported half-lives among other studies may be attributed to differences in mouse strains [33,37,39,41]. Furthermore, Workman et al. observed a longer elimination half-life at 650 mg/kg, attributed to absorption-rate limited elimination and possible saturation of hepatic metabolism [40]. In our study, we also noted a similar trend towards lower clearance at higher doses. While flip flop kinetics or saturable elimination cannot be ruled out, it is important to emphasize that our study employed much lower doses, and this trend lacked significance after excluding one PK sampling time point. The metabolic and excretion pathway of benznidazole is incompletely understood, but it is primarily associated with liver metabolism, with in vivo evidence indicating nitroreduction [40,44–46]. The potential for autoinduction of clearance or absorption mechanisms has previously been discussed, which could impact its disposition [34,47]. However, our study was constrained by the availability of only single-dose data, precluding the exploration of time-dependent pharmacokinetics, which represents a limitation in our research.

A major strength of our study is the application of nonlinear mixed-effects modeling, which allowed us to not only analyze sparse satellite PK data and identify covariate effects, but also to simulate plasma exposures in benznidazole efficacy studies. Thus, ensuring consistency of pharmacokinetics between uninfected mice and those with chronic *T. cruzi* infections is essential. Francisco et al. confirmed similar benznidazole plasma concentrations under various conditions (uninfected mice, acute and chronic infections) [25], supporting the validity of our approach. However, relying on PK data from healthy satellite mice instead of directly measuring plasma exposure in the infected mice used for efficacy studies limits the ability to fully capture inter- and intra-individual variability. Simultaneous assessment of exposure and efficacy in the same infected mice could provide valuable insights into factors influencing treatment outcomes.

## PK/PD relationship for benznidazole

Measuring robust pharmacodynamic endpoints for PK/PD modeling in chronic CD is challenging due to the intermittent detectability of parasites in the bloodstream and their typically rare, transient infection foci [48]. To address this challenge, we derived PD data from a previously published murine model of CD, employing a highly sensitive BLI technique [23,25]. BLI overcomes the limitations of PCR-based methods and offers significant advantages by providing a definite assessment of parasitological cure. Notably, this model accurately predicted the failure of posaconazole to cure human infections, highlighting its potential translational value in CD research [48,49].

In our study, we successfully quantified the exposure-response relationship for benznidazole in *T. cruzi*-infected mice using both univariate and multivariate regression models. $AUC_{\infty}$, $T>IC_{90}$, and $C_{MAX}$ all demonstrated strong associations with the probability of parasitological cure in our dataset, where benznidazole was administered for at least 5 days. However, high collinearity among these predictors hindered the identification of a distinct key driver of benznidazole's antitrypanosomal activity, as indicated by the overlapping 95% CIs of areas under the ROC curves. This limitation of our study underscores the complexity of the dose - response relationship for benznidazole. While the logistic regression models demonstrated high sensitivities, specificities were generally lower, highlighting the challenges in accurately predicting parasitological cure based solely on simulated summary exposure metrics. Incorporating a combination of PK/PD index parameters ($AUC_{\infty}$, $T>IC_{90}$, and $C_{MAX}$) into a multivariate regression model resulted in a small improvement in explained y-variance (goodness-of-fit), suggesting that multiple factors could be relevant for describing benznidazole's

antitrypanosomal efficacy. However, given the limited improvement in classification performance when comparing multivariate with univariate regression, we considered the value of model simplicity for the further evaluation of PK/PD targets.

The $IC_{90}$ used in our analysis was derived from the Tulahuen strain, as comprehensive in vitro data, including protein binding to assay medium, were readily available at the time of the analysis. The $IC_{90}$ for CL Brener amastigotes in assay medium (13.3 μM) is similar to the $IC_{90}$ for the Tulahuen strain (17.4 μM), and relatively small differences in strain susceptibility for benznidazole have been found previously [36,50]. To address uncertainty, we conducted a sensitivity analysis spanning a 10-fold range of $IC_{90}$ values, demonstrating that $T>IC_{90}$ is highly sensitive to $IC_{90}$ variability. This finding underscores the limitations of $T>IC_{90}$ as a PK/PD index parameter and the importance for careful interpretation of PK/PD targets.

Interestingly, simulations revealed that $C_{trough}$ approached zero before the next dose in both once-daily and twice-daily regimens, reflecting the short half-life of benznidazole in mice. This points to a limited relevance of $C_{trough}$ as a predictor of efficacy in this preclinical model. Alternative PK metrics may provide valuable insights for therapeutic drug monitoring in humans, where benznidazole exhibits a longer half-life, but this remains speculative and requires further investigation in clinical studies.

It is worth noting that benznidazole is a prodrug that requires activation by a parasite type I nitroreductase to produce reactive metabolites critical for its antitrypanosomal activity [51,52]. The complexity of the metabolic spectrum and the formation of covalent adducts between benznidazole reduction products and biological molecules [51–55] make it challenging to measure these effects directly at the site of action in the parasite. Consequently, our PK/PD analysis focuses on benznidazole levels in plasma, which serves as a surrogate for the drug's effects but may not fully reflect the activity of the reactive metabolites within the parasite.

A comparison of our study's findings with existing literature is challenging due to the heterogeneity of animal models used in CD research and the lack of standardization, as previously noted [22,56–58]. A significant hurdle is the genetic and phenotypic diversity among *T. cruzi* strains, which exhibit variations in geographic distribution, virulence, disease progression, and drug susceptibility [59,60]. In our study, using the benznidazole - susceptible *T. cruzi* CL Brener strain, chronic infections in BALB/c mice were effectively cured with once daily dosing of 100 mg/kg benznidazole for 5 days [25], or twice daily dosing of 50 mg/kg for 10 days. However, other studies have reported longer treatment durations needed to achieve cure in chronic infection [33,61–64], possibly due to differences in mouse models, timing of treatment initiation, and drug vehicle choice [65]. For example, Cenig et al. found that 100 mg/kg benznidazole administered once daily for 10 days (but not 5 days) cured 100% of BALB/c mice chronically infected with the benznidazole-susceptible Tulahuen strain [64]. Similarly, our univariate logistic regression analysis suggests that achieving PK/PD targets associated with 99% probability of cure may require more than 5 days of once daily dosing with 100 mg/kg benznidazole. A major strength of our study is the integration of efficacy data from an extended dataset and various dosing regimens, thereby allowing for the quantification of dose-exposure-response relationships with enhanced statistical power. Yet, it is important to note that the PK/PD targets identified in our study may not be directly applicable to different experimental conditions. For example, acute infections in the *T. cruzi* CL Brener-BALB/c model require longer treatment durations (20 days, 100 mg/kg) compared to chronic infections [25].

Despite the heterogeneity in experimental models, numerous studies have emphasized the critical role of benznidazole dose and treatment duration in achieving parasitological cure [38,66–69]. For example, Khare et al. reported that during the late acute stage *T. cruzi* CL-infection in mice, a 20-day treatment with 100 mg/kg benznidazole achieved parasitological cure, while dosing for 10 or 15 days resulted in parasitemia rebound [38]. They also observed a dose-dependent increase in antiparasitic activity when comparing benznidazole concentrations of 10, 30, and 100 mg/kg with a 20-day dosing regimen [38]. Likewise, Mazetti et al. reported that the efficacy of benznidazole is dose and time dependent [67]. In their study, both longer treatment durations (up to 40 days at 100 mg/kg benznidazole) and higher doses (ranging from 25 to 300 mg/kg for 20 days) led to improved efficacy in treating acute *T. cruzi* Y-strain and Colombian strain infections in mice [67]. While these studies provide valuable insights for dose optimization, it is crucial to acknowledge the high interdependence among PK/PD index parameters, a challenge we also encountered in our study. Investigating longer treatment

durations at the same dose leads to higher total exposures ($AUC_\infty$) and longer durations above the $IC_{90}$. Vice versa, increasing dose levels while maintaining a constant treatment duration results in higher $C_{MAX}$, $AUC_\infty$ and $T>IC_{90}$. Thus, it is important to recognize that solely altering the dose alone may not definitively pinpoint the primary PD driver of efficacy in the presence of high collinearity among predictors [70]. This complexity is also evident in Molina et al.'s extensive review of benznidazole's efficacy in murine models of CD, revealing a strong correlation between dose - notably both daily and cumulative - and cure across various *T. cruzi* strains [56].

Although a single dose of benznidazole can rapidly reduce parasitaemia by more than 90%, this rapid depletion does not ensure sustained parasite clearance or overall drug efficacy [63,71]. Extended treatment durations are frequently required despite benznidazole's rapid trypanocidal activity [36,63,68,71]. One hypothesis is that spontaneous dormancy contributes to persistent parasite presence and treatment failures [72]. Another hypothesis suggests that parasite replication is an asynchronous process, with various replicative states co-existing within infected cells. The presence of a transient non-replicative state, which is less susceptible to drug-induced toxicity, could ultimately lead to relapse after the successful completion of DNA repair [48,73,74]. Intermittent dosing regimens with higher doses, administered less frequently over an extended period, have been proposed to target replicating parasites and eventually clear the non-replicating states [18,68]. While intermittent dosing demonstrated the potential to also cure hard-to-treat infections [68], treatment outcomes varied and were not necessarily predictable [75]. The effectiveness of this intermittent dosing (once or twice weekly) was attributed to $C_{MAX}$ driving the effect, as opposed to total exposure or sustaining a concentration above a minimum inhibitory concentration (MIC) [18].

Contrary to the strategy of increasing the dose and extending the dosing duration (intermittent dosing) [19,68,75,76], it was also suggested that the standard dose of 5 mg/kg/day benznidazole for 60 days could potentially be reduced to enhance tolerability without compromising efficacy [16,17]. This idea was inspired by the higher efficacy of benznidazole observed in chronically infected children, despite lower plasma concentrations compared to adults [17]. Moreover, population pharmacokinetic modeling and simulation indicated that the commonly accepted target concentrations (3–6 µg/mL) could be achieved with lower doses [16]. However, these suggestions are based on the assumption that maintaining benznidazole levels above the target concentration or $AUC_\infty$ is essential for drug efficacy, which remains uncertain.

Currently, both conflicting approaches are under clinical evaluation, and while they may offer alternative treatment options for CD, their long-term efficacy remains to be shown [15,19,76]. An exploratory analysis comparing simulated plasma concentration - time profiles of benznidazole in mice and humans is shown in (S4 Text).

The analysis suggests that the human standard dosing regimen achieves higher total plasma exposure and a longer duration above the target concentration than the efficacious mouse regimen, but substantially lower peak concentrations. These findings highlight a potential disconnect between exposure levels in mice and humans. However, this remains speculative, as neither the key driver of efficacy nor the optimal human dosing regimen has been established.

Our findings highlight the complexities involved in establishing PK/PD relationships for benznidazole and underscore the need for further research to unravel the interplay among efficacy predictors. The interdependence in PK/PD index parameters can be largely reduced by dose fractionation studies [70], which have been fundamental for the identification of the PK/PD drivers for antibiotic treatments [77]. We can draw valuable lessons from the extensive body of classical literature on this subject [77–80].

## Conclusions

In summary, we have successfully quantified the dose-exposure-response relationship for benznidazole in mice chronically infected with *T. cruzi*. This study represents the first attempt to understand the PK/PD relationship of benznidazole using preclinical BLI imaging data. Given the absence of reliable markers for clinical efficacy or parasitological cure, studying the PK/PD relationship using the BLI model can provide valuable insights and aid in the formulation of relevant clinical hypothesis. Our study underscores the complexity of distinguishing drivers of efficacy due to high collinearity between

PK/PD index parameters. To advance our understanding of factors influencing benznidazole's efficacy, enhanced study designs, such as dose fractionation studies, are proposed.

## Supporting information

**S1 Text. Determination of in vitro antitrypanosomal activity.**
(DOCX)

**S2 Text. Classification performance metrics.**
(DOCX)

**S3 Text. Multivariate analysis.**
(DOCX)

**S4 Text. Simulation of benznidazole exposure in mice vs human.**
(DOCX)

**S1 Fig. Goodness-of-fit for the final population pharmacokinetic model of benznidazole in BALB/c mice.**
(DOCX)

**S2 Fig. Prediction-corrected visual predictive check of the final population pharmacokinetic model for benznidazole.**
(DOCX)

**S3 Fig. Graphical representation of the correlation matrix.**
(DOCX)

**S1 Code. NONMEM code of the final population pharmacokinetic model.**
(DOCX)

**S1 Table. UPLC-MS/MS method.**
(DOCX)

**S2 Table. Secondary pharmacokinetic parameter estimates, based on the final population pharmacokinetic model for benznidazole.**
(DOCX)

**S3 Table. Classification performance for logistic regression models, based on default and optimized probability thresholds.**
(DOCX)

**S4 Table. Sensitivity analysis: $T>IC_{90}$ for dosing regimens in benznidazole efficacy studies, based on a range of $IC_{90}$ values.**
(DOCX)

**S5 Table. Sensitivity analysis: univariate logistic regression diagnostics, based on a range of $IC_{90}$ values.**
(DOCX)

## Acknowledgments

The authors thank Sergey Kucheryavski for his scientific advice on multivariate analysis, particularly regarding the use of the R package mdatools. Additionally, we are grateful to Sue J. Lee and Mavuto Mukaka for their insightful discussions.

## Author contributions

**Conceptualization:** Frauke Assmus, Ayorinde Adehin, Richard M Hoglund, Ivan Scandale, Joel Tarning.

**Data curation:** Frauke Assmus, Ayorinde Adehin.

**Formal analysis:** Frauke Assmus, Ayorinde Adehin, Richard M Hoglund, Amanda Fortes Francisco, Joel Tarning.

**Funding acquisition:** Eric Chatelain, Ivan Scandale, Joel Tarning.

**Investigation:** Frauke Assmus, Ayorinde Adehin, Richard M Hoglund, Amanda Fortes Francisco, David M Shackleford, Ivan Scandale, Joel Tarning.

**Methodology:** Frauke Assmus, Ayorinde Adehin, Richard M Hoglund, Amanda Fortes Francisco, John M Kelly, Joel Tarning.

**Project administration:** Richard M Hoglund, Susan A. Charman, Karen L White, Fanny Escudié, Eric Chatelain, Ivan Scandale, Joel Tarning.

**Resources:** Eric Chatelain, Ivan Scandale, Joel Tarning.

**Software:** Richard M Hoglund, Joel Tarning.

**Supervision:** Richard M Hoglund, John M Kelly, Susan A. Charman, Fanny Escudié, Eric Chatelain, Ivan Scandale, Joel Tarning.

**Validation:** Frauke Assmus, Ayorinde Adehin, Richard M Hoglund, Amanda Fortes Francisco, Joel Tarning.

**Visualization:** Frauke Assmus, Ayorinde Adehin, Richard M Hoglund, Joel Tarning.

**Writing – original draft:** Frauke Assmus.

**Writing – review & editing:** Frauke Assmus, Ayorinde Adehin, Richard M Hoglund, Amanda Fortes Francisco, Michael D Lewis, John M Kelly, Susan A. Charman, Karen L White, David M Shackleford, Fanny Escudié, Eric Chatelain, Ivan Scandale, Joel Tarning.

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
