## [Decision Letter · Decision Letter 0]

6 Dec 2024

PNTD-D-24-01371

Pharmacokinetic-pharmacodynamic analysis of benznidazole in a murine model of chronic Chagas disease

Dear Dr. Tarning,

Thank you for submitting your manuscript to PLOS Neglected Tropical Diseases. After careful consideration, we feel that it has merit but does not fully meet PLOS Neglected Tropical Diseases's publication criteria as it currently stands. Therefore, we invite you to submit a revised version of the manuscript that addresses the points raised during the review process.

Please submit your revised manuscript within 60 days Feb 04 2025 11:59PM. If you will need more time than this to complete your revisions, please reply to this message or contact the journal office at plosntds@plos.org. Please include the following items when submitting your revised manuscript:

We look forward to receiving your revised manuscript.

Kind regards,

Elham Kazemirad 

Academic EditorPPLOS Neglected Tropical Diseases

Guilherme Werneck

Section Editor

Shaden Kamhawi

co-Editor-in-Chief

Paul Brindley

co-Editor-in-Chief

**Journal Requirements:**

3) Please amend your detailed Financial Disclosure statement. This is published with the article. It must therefore be completed in full sentences and contain the exact wording you wish to be published. Please ensure that the funders and grant numbers match between the Financial Disclosure field and the Funding Information tab in your submission form. Note that the funders must be provided in the same order in both places as well.

**Reviewers' Comments:**

Reviewer's Responses to Questions

**Key Review Criteria Required for Acceptance?**

**Methods**

-Are the objectives of the study clearly articulated with a clear testable hypothesis stated?

-Is the study design appropriate to address the stated objectives?

-Is the population clearly described and appropriate for the hypothesis being tested?

-Is the sample size sufficient to ensure adequate power to address the hypothesis being tested?

-Were correct statistical analysis used to support conclusions?

-Are there concerns about ethical or regulatory requirements being met?

Reviewer #1: Yes, the design and methods are appropriate. I have one (probably minor) concern: The BLI model is done with the CL Brener strain, but the IC90 for their PK analysis was done using the Tulahuen strain. I wonder why they don't use the IC90 from the CL Brener strain?

Reviewer #2: - How is chronic infection defined in this mouse model?

- what was the rationale for the (varying) sample sizes for each treatment dose / duration? Animal numbers per group seem very different from one another (e.g. how were sample sizes defined / calculated? )

- “The final population PK model for benznidazole in satellite mice…” What are “satellite mice”?

- I understand that there were 92 plasma samples included in the PK analysis? From 118 mice? Is that correct? Were there multiple samples from some mice and none from others? Maybe, also add this information to Fig 1 too (i.e. total number of benznidazole measurements in the model)

Reviewer #3: please refer to the general comments

**Results**

-Does the analysis presented match the analysis plan?

-Are the results clearly and completely presented?

-Are the figures (Tables, Images) of sufficient quality for clarity?

Reviewer #1: Results are clearly presented.

Reviewer #2: - “parasitological cure was achieved in over 90% of chronically infected mice when treated once daily with 100 mg/kg benznidazole for 5 or 10 days (scenario a and b), or twice daily with 50 mg/kg benznidazole for 10 days (scenario c). In contrast, once daily dosing with 50 mg/kg benznidazole for 10 days (scenario d) cured only 66.7% of infected mice”. A measure of uncertainty of these estimates should be included (e.g. 95% CI). The way this is written, it seems to imply that 50 mg/kg BID x 10 days was clearly different from 50 mg/kg OD x 10 days, whereas the first group only had 3 mice in it. At least add the N of each group in the text (so that the reader doesn’t have to go back and forth to the table all the time to understand the text).

- Table 1 (and every time that the dosing schedules are mentioned): When (e.g.)50 mg/kg BID is mentioned, do the authors mean that each time (in this case twice per day) the mice received a 50 mg/kg dose (i.e. a 100 mg/kg/day dose), or a 50 mg/kg /day dose in 2 administrations (i.e. 25 mg/kg twice a day)?

- “In the present study, a higher dose was associated with a significantly slower absorption rate (ΔOFV = -31.3), as indicated by a decrease in KA (power function). At the lowest, medium and highest dose, KA was estimated as 5.11, 2.18 and 0.86 h-1, respectively.” Please add some estimate of uncertainty around the point estimates. Also, were the higher doses administered in higher liquid volumes?

- ” Goodness-of-fit diagnostics (AIC, BIC, Mc Fadden R2) indicated slightly stronger explanatory power for AUC∞ in binary regression compared to CMAX”. How precise is the estimation of CMAX in a model with such sparse sampling? Also, wouldn’t cumulative dose be a better explanatory variable? Was this tested?

- Multivariate analysis: similar to above, was cumulative dose included in the analysis?

Reviewer #3: please refer to the general comments

**Conclusions**

-Are the conclusions supported by the data presented?

-Are the limitations of analysis clearly described?

-Do the authors discuss how these data can be helpful to advance our understanding of the topic under study?

-Is public health relevance addressed?

Reviewer #1: The conclusions are supported by the data.

Reviewer #2: - “Administering benznidazole as a suspension at doses of 30 mg/kg and 100 mg/kg resulted in delayed absorption (TMAX = 0.86 h and 1.47 h, respectively) compared to the clear 10 mg/kg solution (TMAX = 0.49 h).” As suggested above, some detail on actual administrated liquid volumes would be useful here

- “we also noted a similar trend towards lower clearance at higher doses. While flip flop kinetics or saturable elimination cannot be ruled out” What about enterohepatic circulation?

Reviewer #3: please refer to the general comments

**Editorial and Data Presentation Modifications?**

Reviewer #1: No comments

Reviewer #2: The title could be a bit more clear on what the authors mean as pharmacodynamics in their experimental setting.

It is important to define what “PD” means. In this case, it seems that the authors refer to drug response (defined as absence of bioluminescence after immunosuppression with cyclophosphamide, I guess).

Maybe the title should add information on this (e.g. “Pharmacokinetic-pharmacodynamic analysis of T Cruzi elimination after benznidazole treatment in a murine model of chronic Chagas disease”

It may also be worth considering (and clarifying) that this is not really a murine model of chronic chagas disease (e.g. there’s no real chronic cardiac or Gi pathology similar to what humans develop, etc). This is a murine model of chronic T cruzi infection.

Reviewer #3: please refer to the general comments

**Summary and General Comments**

Reviewer #1: The manuscript by Assmus et al describes a PK-PD analysis of the antiparasitic drug, benznidazole, in the murine model of chronic T. cruzi infection. This work is justified since benznidazole remains the primary therapeutic for treating T. cruzi infection in humans. Despite it's long use since the 1970s, there remain many questions about the optimal dosing and duration of therapy to benefit chronically infected patients. In fact there are ongoing clinical trials to test shorter courses and different doses mainly to improve the poor side-effect profile while maintaining comparable efficacy. The research in this paper can theoretically help guide the clinical use of benznidazole to reproduce the PK-PD profile that is effective in mice.

For these studies, the investigators relied on data generated with the bioluminescence model of T. cruzi infection. This is the most robust animal model for determining parasitological cure of the animals. The results of using benznidazole treatment in the chronic infection model had been generated previously. The PK data with benznidazole was obtained with varying dosing regimens in uninfected mice of the same strain. Their PK results were pretty much consistent with published data. The analysis of the exposure-response relationship showed that AUC, T>IC90, and Cmax ALL demonstrated strong associations with the probability of parasitological cure when benznidazole was administered for at least five days. The high collinearity among those parameters prevented the identification of a distinct PK DRIVER of benznidazole's antitrypanosomal activity. As pointed out by the authors, detailed dose fractionation studies may be needed to better answer the fundamental question about the PK driver for this drug. They propose doing that work in future studies.

Although the studies don't lead to decisive conclusion about the exposure-response relationship between benznidazole and antiparasitic activity in chronic murine T. cruzi infection, the data do expand our general understanding of this relationship and make clear there's not a real simple answer. The authors have drawn reasonable conclusions based on the presented data. This research is a helpful addition to the Chagas disease literature. My only comment for attention was listed above about the Methods.

Reviewer #2: This is an interesting animal PK-PD model. It adds some insight to the literature available, and attempts to link the observations to human observations (e.g. in S4) at least to a limited extent. Some of the commnents made in S4 could be in the actual discussion to help readers link the mouse model to what actually goes on in humans.

Reviewer #3: The study by Assmus et al. investigated the quantitative PK/PD relationship of benznidazole in a murine model of chronic Chagas disease using the population PK approach. This represents the first of its kind using efficacy data generated from the BLI imaging data. The described work is highly relevant to the scope of PLOS-NTD and advances our understanding of factors driving benznidazole efficacy in chronic Chagas diseases. In general, the manuscript has been clearly written and organized with sufficient details on the methodologies employed and the results presented. Discussion was found to be pertinent and intriguing, and conclusions were supported by the study results. However, several moderate/minor concerns should be addressed prior to acceptance.

1. Authors should use the opportunity to assess factors that both drive efficacy and could potentially be employed for future therapeutic drug monitoring (TDM) purposes. Currently, Cmax, AUC and T>IC90 have been identified as the driving factors, however these measurements will be impossible for use in clinics to monitor treatment and predict efficacy. More realistic ones are C_trough values (e.g., at 12h, 24h or before last dose). It’d be very interesting to know whether these factors drive efficacy as well.

2. Authors should also acknowledge the limitation of using satellite PK from healthy mice rather than PK/exposure from the infected mice used in the efficacy study. It was mentioned (Line 526) that PK from uninfected and infected mice were similar, which is important. But this does not take away the additional insights one can get from assessing exposure and efficacy on the same subject, due to interindividual and intraindividual variability and ability to identify factors that drive efficacy on the individual level and can be potentially used for TDM.

3. Line 153, please provide the dose formulation concentrations (e.g., mg/ml) used for different dose regimens, as it was reasoned later (Line 500) that differences in the dose suspension concentration may have led to difference in absorption kinetics.

4. Line 157, why was food kept from mice for this group?

5. Line 198, add refs for these allometric scaling factors.

6. Remove the strikethrough on line 181 in SI.

7. Fig 1: it’s unclear what type of observed data are plotted here. Are they from the same dose regimen group? Which one?

PLOS authors have the option to publish the peer review history of their article (what does this mean? ). If published, this will include your full peer review and any attached files.

**Do you want your identity to be public for this peer review?** For information about this choice, including consent withdrawal, please see our Privacy Policy .

Reviewer #1: No

Reviewer #2: No

Reviewer #3: No

**Figure resubmission:**
---

## [Decision Letter · Decision Letter 1]

16 Jan 2025

PNTD-D-24-01371R1Pharmacokinetic-pharmacodynamic modeling of benznidazole and its antitrypanosomal activity in a murine model of chronic Chagas diseasePLOS Neglected Tropical Diseases  Dear Dr. Tarning,

Thank you for submitting your manuscript to PLOS Neglected Tropical Diseases. After careful consideration, we feel that it has merit but does not fully meet PLOS Neglected Tropical Diseases's publication criteria as it currently stands. Therefore, we invite you to submit a revised version of the manuscript that addresses the points raised during the review process.

Please submit your revised manuscript within 30 days Feb 15 2025 11:59PM. If you will need more time than this to complete your revisions, please reply to this message or contact the journal office at plosntds@plos.org. Please include the following items when submitting your revised manuscript:

* A rebuttal letter that responds to each point raised by the editor and reviewer(s). You should upload this letter as a separate file labeled 'Response to Reviewers '. This file does not need to include responses to any formatting updates and technical items listed in the 'Journal Requirements' section below. * A marked-up copy of your manuscript that highlights changes made to the original version. You should upload this as a separate file labeled 'Revised Manuscript with Track Changes '. * An unmarked version of your revised paper without tracked changes. You should upload this as a separate file labeled 'Manuscript '. If you would like to make changes to your financial disclosure, competing interests statement, or data availability statement, please make these updates within the submission form at the time of resubmission. Guidelines for resubmitting your figure files are available below the reviewer comments at the end of this letter. We look forward to receiving your revised manuscript. Kind regards, Elham Kazemirad, Ph.DAcademic EditorPLOS Neglected Tropical Diseases Guilherme WerneckSection EditorPLOS Neglected Tropical Diseases

Shaden Kamhawi

co-Editor-in-Chief

Paul Brindley

co-Editor-in-Chief

**Additional Editor Comments:** Please check Reviewer #3 comments on Figure 1**Journal Requirements:**

Please ensure that the funders and grant numbers match between the Financial Disclosure field and the Funding Information tab in your submission form. Note that the funders must be provided in the same order in both places as well. State the initials, alongside each funding source, of each author to receive each grant. For example: "This work was supported by the National Institutes of Health (####### to AM; ###### to CJ) and the National Science Foundation (###### to AM).".

**Reviewers' comments:** Reviewer's Responses to Questions

**Key Review Criteria Required for Acceptance?**

**Methods**

-Are the objectives of the study clearly articulated with a clear testable hypothesis stated?

-Is the study design appropriate to address the stated objectives?

-Is the population clearly described and appropriate for the hypothesis being tested?

-Is the sample size sufficient to ensure adequate power to address the hypothesis being tested?

-Were correct statistical analysis used to support conclusions?

-Are there concerns about ethical or regulatory requirements being met?

Reviewer #1: This is a revised manuscript. The authors have adequately addressed the concerns raised in my initial review (Reviewer 1).

Reviewer #2: the authors have addressed all my questions

Reviewer #3: (No Response)

**Results**

-Does the analysis presented match the analysis plan?

-Are the results clearly and completely presented?

-Are the figures (Tables, Images) of sufficient quality for clarity?

Reviewer #1: This is a revised manuscript. The authors have adequately addressed the concerns raised in my initial review (Reviewer 1).

Reviewer #2: the authors have addressed all my questions

Reviewer #3: (No Response)

**Conclusions**

-Are the conclusions supported by the data presented?

-Are the limitations of analysis clearly described?

-Do the authors discuss how these data can be helpful to advance our understanding of the topic under study?

-Is public health relevance addressed?

Reviewer #1: This is a revised manuscript. The authors have adequately addressed the concerns raised in my initial review (Reviewer 1).

Reviewer #2: the authors have addressed all my questions

Reviewer #3: (No Response)

**Editorial and Data Presentation Modifications?**

Reviewer #1: Accept

Reviewer #2: the authors have addressed all my questions

Reviewer #3: (No Response)

**Summary and General Comments**

Reviewer #1: This is a revised manuscript. The authors have adequately addressed the concerns raised in my initial review (Reviewer 1).

Reviewer #2: the authors have addressed all my questions

Reviewer #3: Authors addressed all my previous concerns, except for the confusion on Figure 1. The way the Fig. 1 is currently presented does not serve the purpose of visual predictive check of the pop-pk model. What’s the point of averaging benznidazole plasma concentrations from all three dose groups (10, 30 and 100 mpk), i.e., the solid red line? Plasma concentrations are supposed to be different from different doses. The shaded areas do not make sense either. What dose was used in the model-simulation to generate these? A reasonable way is to delineate the data among different dose groups using different symbols, plot median and 5th and 95th percentiles of the data for each dose group, and simulate 95% CI with shaded areas for specific dose. This way will allow visual check of overlap between the data and simulation.

PLOS authors have the option to publish the peer review history of their article (what does this mean? ). If published, this will include your full peer review and any attached files.

**Do you want your identity to be public for this peer review?** For information about this choice, including consent withdrawal, please see our Privacy Policy .

Reviewer #1: No

Reviewer #2: **Yes: ** Facundo Garcia-Bournissen

Reviewer #3: No

**Figure resubmission:** While revising your submission, please upload your figure files to the Preflight Analysis and Conversion Engine (PACE) digital diagnostic tool, https://pacev2.apexcovantage.com/. PACE helps ensure that figures meet PLOS requirements. To use PACE, you must first register as a user. Registration is free. Then, login and navigate to the UPLOAD tab, where you will find detailed instructions on how to use the tool. If you encounter any issues or have any questions when using PACE, please email PLOS at figures@plos.org. Please note that Supporting Information files do not need this step. If there are other versions of figure files still present in your submission file inventory at resubmission, please replace them with the PACE-processed versions.**Reproducibility:** To enhance the reproducibility of your results, we recommend that authors of applicable studies deposit laboratory protocols in protocols.io, where a protocol can be assigned its own identifier (DOI) such that it can be cited independently in the future. Additionally, PLOS ONE offers an option to publish peer-reviewed clinical study protocols. Read more information on sharing protocols at https://plos.org/protocols?utm_medium=editorial-email&utm_source=authorletters&utm_campaign=protocols

---

## [Decision Letter · Decision Letter 2]

20 Feb 2025

PNTD-D-24-01371R2Pharmacokinetic-pharmacodynamic modeling of benznidazole and its antitrypanosomal activity in a murine model of chronic Chagas diseasePLOS Neglected Tropical DiseasesDear Dr. Tarning, Thank you for submitting your manuscript to PLOS Neglected Tropical Diseases. After careful consideration, we feel that it has merit but does not fully meet PLOS Neglected Tropical Diseases's publication criteria as it currently stands. Therefore, we invite you to submit a revised version of the manuscript that addresses the points raised during the review process. Please submit your revised manuscript within 30 days Mar 22 2025 11:59PM. If you will need more time than this to complete your revisions, please reply to this message or contact the journal office at plosntds@plos.org. Please include the following items when submitting your revised manuscript: * A rebuttal letter that responds to each point raised by the editor and reviewer(s). You should upload this letter as a separate file labeled 'Response to Reviewers '. This file does not need to include responses to any formatting updates and technical items listed in the 'Journal Requirements' section below. * A marked-up copy of your manuscript that highlights changes made to the original version. You should upload this as a separate file labeled 'Revised Manuscript with Track Changes '. * An unmarked version of your revised paper without tracked changes. You should upload this as a separate file labeled 'Manuscript '. If you would like to make changes to your financial disclosure, competing interests statement, or data availability statement, please make these updates within the submission form at the time of resubmission. Guidelines for resubmitting your figure files are available below the reviewer comments at the end of this letter. We look forward to receiving your revised manuscript. Kind regards, Elham Kazemirad, Ph.DAcademic EditorPLOS Neglected Tropical Diseases Guilherme WerneckSection EditorPLOS Neglected Tropical Diseases

Shaden Kamhawi

co-Editor-in-Chief

Paul Brindley

co-Editor-in-Chief

**Additional Editor Comments:**

Please prepare the all Tables and Figures based on the PLOS style. Please edit the references and add doi.Check the Submission Guidelines.**Reviewers' comments:** Reviewer's Responses to Questions

**Key Review Criteria Required for Acceptance?**

**Methods** :

-Are the objectives of the study clearly articulated with a clear testable hypothesis stated?

-Is the study design appropriate to address the stated objectives?

-Is the population clearly described and appropriate for the hypothesis being tested?

-Is the sample size sufficient to ensure adequate power to address the hypothesis being tested?

-Were correct statistical analysis used to support conclusions?

-Are there concerns about ethical or regulatory requirements being met?

Reviewer #3: (No Response)

**Results** :

-Does the analysis presented match the analysis plan?

-Are the results clearly and completely presented?

-Are the figures (Tables, Images) of sufficient quality for clarity?

Reviewer #3: (No Response)

**Conclusions** :

-Are the conclusions supported by the data presented?

-Are the limitations of analysis clearly described?

-Do the authors discuss how these data can be helpful to advance our understanding of the topic under study?

-Is public health relevance addressed?

Reviewer #3: (No Response)

**Editorial and Data Presentation Modifications?**

Reviewer #3: (No Response)

**Summary and General Comments** :

Reviewer #3: I appreciate the addition of Fig S2 for transparency. Although upon closer examination, most of observed data (open circles) in Fig 1 and Fig S2 do not match each other. Please have them checked and corrected.

In addition, please prepare and include a data table in the supplemental information to show all the time and plasma concentration values for the observed data.

PLOS authors have the option to publish the peer review history of their article (what does this mean? ). If published, this will include your full peer review and any attached files.

**Do you want your identity to be public for this peer review?** For information about this choice, including consent withdrawal, please see our Privacy Policy .

Reviewer #3: No

---

## [Decision Letter · Decision Letter 3]

7 Mar 2025

Dear Professor Tarning,

We are pleased to inform you that your manuscript 'Pharmacokinetic-pharmacodynamic modeling of benznidazole and its antitrypanosomal activity in a murine model of chronic Chagas disease' has been provisionally accepted for publication in PLOS Neglected Tropical Diseases.

Best regards,

Elham Kazemirad, Ph.D

Academic Editor

Guilherme Werneck

Section Editor

Shaden Kamhawi

co-Editor-in-Chief

Paul Brindley

co-Editor-in-Chief

Reviewer's Responses to Questions

**Key Review Criteria Required for Acceptance?**

**Methods**

-Are the objectives of the study clearly articulated with a clear testable hypothesis stated?

-Is the study design appropriate to address the stated objectives?

-Is the population clearly described and appropriate for the hypothesis being tested?

-Is the sample size sufficient to ensure adequate power to address the hypothesis being tested?

-Were correct statistical analysis used to support conclusions?

-Are there concerns about ethical or regulatory requirements being met?

Reviewer #3: (No Response)

**Results**

-Does the analysis presented match the analysis plan?

-Are the results clearly and completely presented?

-Are the figures (Tables, Images) of sufficient quality for clarity?

Reviewer #3: (No Response)

**Conclusions**

-Are the conclusions supported by the data presented?

-Are the limitations of analysis clearly described?

-Do the authors discuss how these data can be helpful to advance our understanding of the topic under study?

-Is public health relevance addressed?

Reviewer #3: (No Response)

**Editorial and Data Presentation Modifications?**

Reviewer #3: (No Response)

**Summary and General Comments**

Reviewer #3: Thanks for the clarification. I have no further comments.

PLOS authors have the option to publish the peer review history of their article (what does this mean? ). If published, this will include your full peer review and any attached files.

**Do you want your identity to be public for this peer review?** For information about this choice, including consent withdrawal, please see our Privacy Policy .

Reviewer #3: No

---

## [Editor Report · Acceptance letter]

Dear Professor Tarning,

We are delighted to inform you that your manuscript, "Pharmacokinetic-pharmacodynamic modeling of benznidazole and its antitrypanosomal activity in a murine model of chronic Chagas disease," has been formally accepted for publication in PLOS Neglected Tropical Diseases.

Best regards,

Shaden Kamhawi

co-Editor-in-Chief

Paul Brindley

co-Editor-in-Chief
